# An Overview of Wearable Haptic Technologies and Their Performance in Virtual Object Exploration

**DOI:** 10.3390/s23031563

**Published:** 2023-02-01

**Authors:** Myla van Wegen, Just L. Herder, Rolf Adelsberger, Manuela Pastore-Wapp, Erwin E. H. van Wegen, Stephan Bohlhalter, Tobias Nef, Paul Krack, Tim Vanbellingen

**Affiliations:** 1Department of Precision and Microsystems Engineering, Delft University of Technology, 2628 CD Delft, The Netherlands; 2Sensoryx AG, 8005 Zurich, Switzerland; 3Gerontechnology and Rehabilitation Group, ARTORG Center for Biomedical Engineering Research, University of Bern, 3008 Bern, Switzerland; 4Neurocenter, Luzerner Kantonsspital, 6000 Luzern, Switzerland; 5Department of Rehabilitation Medicine, Amsterdam Movement Sciences, Amsterdam UMC, VUmc, 1117 HV Amsterdam, The Netherlands; 6Department of Neurology, Center for Parkinson’s Disease and Movement Disorders, Inselspital, Bern University Hospital, University of Bern, 3008 Bern, Switzerland

**Keywords:** object exploration, object interaction, virtual reality, haptics, wearable, VR system, overview

## Abstract

We often interact with our environment through manual handling of objects and exploration of their properties. Object properties (OP), such as texture, stiffness, size, shape, temperature, weight, and orientation provide necessary information to successfully perform interactions. The human haptic perception system plays a key role in this. As virtual reality (VR) has been a growing field of interest with many applications, adding haptic feedback to virtual experiences is another step towards more realistic virtual interactions. However, integrating haptics in a realistic manner, requires complex technological solutions and actual user-testing in virtual environments (VEs) for verification. This review provides a comprehensive overview of recent wearable haptic devices (HDs) categorized by the OP exploration for which they have been verified in a VE. We found 13 studies which specifically addressed user-testing of wearable HDs in healthy subjects. We map and discuss the different technological solutions for different OP exploration which are useful for the design of future haptic object interactions in VR, and provide future recommendations.

## 1. Introduction

Virtual reality (VR) has recently emerged in a wide variety of applications. In the field of games, rehabilitation, medicine, and sports, the benefits of such VEs have shown their value [1]. Safety simulation training in VR is used to improve performance in, for example, the maritime sector [2], and surgical skills can be trained with VR simulation systems to enhance surgical performance in real life [3]. In the case of rehabilitation, VR is used to enhance patient engagement, while also relieving work pressure for professional staff, such as physiotherapists, while providing more flexibility in creating patient-tailored rehabilitation programs [4]. Even though VR interactions with auditory and visual feedback only are successfully applied in rehabilitation settings [5,6], implementing haptic feedback can further enhance the interaction, immersion, and imagination [7] in a VE. The challenge, however, lies in the actual implementation of the haptic feedback in a realistic manner. 

In the physical world, we are constantly stimulated by the environment by complex multi-modal haptic stimuli that we perceive with our haptic perception system. This system, also known as the somatosensory system, can be physiologically classified into kinaesthetic and tactile perception based on the location of the sensory receptors. Tactile perception relies on cutaneous receptors in the skin that can perceive mechanical stimuli, such as high/low frequency vibrations, pressure and shear deformation, as well as electrical stimuli, temperature and chemicals. Kinaesthetic perception relies on sensory receptors in muscles, tendons, and joints that reflect the operational state of the human locomotor system, such as joint positions, limb alignment, body orientation and muscle tension [8,9,10]

By interacting with objects and by manually exploring their intrinsic and extrinsic properties, such as texture, stiffness, size, shape, temperature, weight, and orientation, we can perceive and interact with our environment. To re-create such extensive haptic experiences in a virtual setting requires complex technological solutions and this is a fast-growing area of interest for researchers and engineers. Because we often interact with the environment using our hands, much of the focus in haptic technology research has been dedicated to hand-based devices. Ungrounded devices, i.e., with no external frame of reference, in particular, have been developed due to their advantages in flexibility and range of motion. These devices can be further categorized into different types: wearable devices, such as fingertip, glove-based or exoskeleton devices, or handheld and tool-based devices. The latter have the advantage that they can be directly used by grasping, without the device having to be attached to the users’ hand. Wearables (see Figure 1), on the other hand, allow more freedom and movement of the users’ hand and fingers while interacting with a VE. They create a more immersive and realistic feeling and, therefore, seem promising for the improvement of virtual experience [11,12].

There is a huge interest in the design and development of object-exploration-oriented haptic technologies [13,14,15,16], and several studies have already stressed the importance of wearable haptic technologies [17,18,19,20,21]. However, many of the available technologies have not been tested for their applicability and performance in a VE. As far as we know, there is no overview of HDs in which current wearable haptic technologies are considered against VR user-verified object-exploration tasks. 

## 2. State of the Art/Aim

This study aims to provide a new comprehensive review of the state of the art, and of the relevant design features of recent wearable HDs that have been verified in user experiments, by categorizing them based on their intended OP exploration. Such a review can be useful in understanding how certain design solutions are related to exploration of virtual OP and how this affects user performance and experience.

## 3. Materials and Methods

### 3.1. Search Strategy

For this overview, we focused on including papers which were published between 2015 and October 2022, and used relevant databases, such as PubMed and IEEE Explore, together with an extensive Google Scholar search. The main keywords for the search strategy were: haptic, cutaneous, tactile, kinaesthetic, force feedback AND object, interaction, shape, stiffness, texture, weight, thermal, orientation AND virtual, augmented reality AND wearable, glove, exoskeleton, fingertip device, AND subjects, volunteers, participants, users. First, all titles were screened and, where deemed necessary, abstracts were reviewed. Then, all relevant original research articles were examined in detail, including a review of the references in each publication to identify additional sources.

### 3.2. Eligibility Criteria (Inclusion and Exclusion) and Study Selection

The scope of the studies included on HDs is visually depicted in Figure 2. The selection criteria included that the study should, firstly, concern the development and testing of a wearable hand-based HD (i.e., a glove, exoskeleton, or fingertip device) that does not restrict the user. Secondly, the study should contain an experimental part where the developed device is tested in a VE, augmented reality (AR) environment or a remote environment (RE) with healthy participants. The test should be based on an intrinsic or extrinsic OP exploration task (i.e., texture, compliance, thermal quality, shape, size, weight, orientation). No restrictions were made on the type and set-up of the experiment. Two reviewers (MvW, TV) independently screened all titles and abstracts for the eligibility criteria.

### 3.3. Overview Layout

All the included studies were sorted per object property and several characteristics were collected from the used devices. The technical properties of the device included the type of haptic feedback, the haptic technology and the tracking technology. For performance testing of the devices, the exact experiment, the results, the conclusion, and recommendations for future work were collected.

## 4. Results

A flowchart of the results of the selection process is shown in Figure 3. After preliminary screening, the records that were discarded did not contain an actual user test in a VE that addressed OP exploration or the device was deemed as not wearable. After this screening, three full articles were discarded after reading the complete article, due to the actual testing not fitting into one of the OP exploration modalities. 

An overview of the resulting included HDs can be found in Table 1, sorted by their specific OP exploration task for which they were tested in a VE, AR environment or RE For each HD, the function states the OP exploration modality; the type of feedback is described as either cutaneous or kinaesthetic. Furthermore, the implemented haptic technology is briefly described, as well as the tracking technology and a short summary of the conclusions on the performance of the device in the VR OP exploration user testing. The contents of Table 1 are described in more detail in Section 4.1, Section 4.2, Section 4.3, Section 4.4, Section 4.5 and Section 4.6, where they are sorted according to their function. 

### 4.1. Texture Exploration

Keef et al. [22] designed a glove to simulate binary gradations of the sensation of roughness through an electrotactile signal from a specifically synthesized conductive, bio-inspired pi-conjugated elastomer. Both trained and untrained participants using the VR system were able to distinguish rough versus smooth surfaces. Hardness was simulated through eccentric rotating mass (ERM) vibration actuators on each fingertip and the palm at a frequency of 60 Hz and varying amplitudes, where the amplitude of the vibration was greater for “hard” sensations. The accuracy for distinguishing hardness was an average of 90% for trained and untrained individuals. Roughness was simulated by an electrotactile signal through a conductive elastomer, where a smooth surface was simulated as a continuous signal and a rough or bumpy surface as an intermittent signal. The accuracy for roughness was an average of 85% for both trained and untrained individuals. Trained participants were better at identifying the intended tactile sensations. Future work should focus on the psychophysical aspects in haptic feedback, such as determining perceptual thresholds and confusion of sensations. 

Yem et al. [10] developed a fingertip device, FinGAR, that can give four dimensions of tactile feedback. It has a direct current (DC) motor that provides high-frequency vibrations and shear deformation together with an array of electrodes with high spatial resolution to provide pressure and low-frequency vibrations through cathodic and anodic stimulation on the thumb, index, and middle finger. Ten participants had to answer questions about the perception of virtual roughness, friction, and hardness of objects after exploring them in VR using a mouse and wearing the device for feedback. Macro roughness was best perceived through skin deformation and cathodic stimulation, friction and fine roughness were best perceived through high frequency vibration, and hardness was best perceived through skin deformation. Future work should focus on accurate control of the intensities of the four tactile dimensions. 

### 4.2. Thermal Quality Exploration

Li S et al. [23] created a glove that transmits multi-modal tactile information through miniature ERM actuators and two semiconductor refrigerators. They compared a real-life experiment to a VR experiment where two chemicals underwent endothermic and exothermic reactions. The temperature change was perceived by the subjects both in real life and in VR. All subjects were able to define an exothermic experiment in VR as heating and almost all subjects were able to perceive an endothermic reaction as cooling. They also tested the heat perception of different temperature rising intensities and found that the subjects were most sensitive to slow and fast heating, but not to intermediate temperatures. Training had no significant effect on recognition accuracy. 

The glove from Keef et al. [22] described in Section 3.1 also contained miniature thermo-electric devices to simulate warm and cool surface temperatures in the same experiment. The results showed 100% accuracy for distinguishing binary gradations of warm and cool sensation of the panels. However, the cool side heated up after about 5s, making it necessary to include better heat management for future studies, for example, by using heat sinks.

### 4.3. Compliance Exploration

Lee et al. [24] designed a wearable HD that adopts passive actuation with a tendon-based transmission mechanism to generate kinaesthetic feedback to the thumb and index finger for three different stiffness modes. Subjects were asked to compare the stiffness of real-world objects to that of objects in the virtual world while receiving different types of haptic feedback: rigid kinaesthetic, elastic kinaesthetic, vibrotactile haptic and pseudo haptic, and no haptic feedback. Results were collected through multiple-choice questions. Kinaesthetic feedback was most effective in rendering rigid objects, giving the highest realness values. The results verified that the mechanism can render rigid and elastic/deformable virtual objects. The authors concluded that vibrotactile feedback is similarly effective in rendering elasticity, but the realness of the virtual object is higher under the kinaesthetic feedback conditions. Moreover, they confirmed the importance of proper visual haptic feedback in finger-based manipulation. 

Mo et al. [25] developed a fingertip device with three small servo motors and an uncoupled five-bar and a slider-crank linkage combination to provide 3-DoF force feedback. Subjects (*n* = 20) tried out the device before the start of the experiment. During the experiment the subjects were asked to discriminate between three groups of three springs with equal spring difference within a group but varying in stiffness difference per group. The results showed that, as the spring difference got smaller, the subjects had more difficulties in distinguishing the stiffnesses. Moreover, the correctness of the discrimination was related to the absolute value of the stiffness, suggesting that greater stimulus differences were required to perceive the same difference when the stiffness increased. Overall, the devices proved to be capable of rendering variable stiffnesses.

Hosseini et al. [26] built a haptic exoskeleton based on force feedback by means of a twisted string actuation (TSA) system to compare a virtual spring stiffness with that of a real spring between the thumb and index finger. The TSA system consisted of tendons to connect each finger and the thumb to two small DC motors, which provided force feedback linear to the displacement of the virtual spring times the spring constant. The results of the experimental study with eight healthy participants showed that the participants were better at recognizing a difference in spring stiffness between the real and virtual spring when both the glove and the visuals conveyed different stiffnesses, compared to when the glove would not convey a different stiffness. Moreover, a higher spring stiffness resulted in better accuracy and adding correct visual feedback greatly enhanced the accuracy of the task. The results suggested that the users rely greatly on visual cues next to the haptic information in detecting spring stiffness.

Maereg et al. [27] designed a wearable HD with vibrotactile feedback by means of five ERM actuators to discriminate different values of stiffness in VR. They performed an experiment with ten subjects that had to perceive the stiffness of a virtual spring with only visual feedback, only tactile feedback and with a combination of both. The virtual force exerted by the spring on the users’ hand during collision was communicated through the vibration actuators, with strength of the vibration’s stiffness values being proportional to the spring forces. They found that the tactile feedback only showed better performance on higher stiffness values, whereas the tactile only and combined feedback resulted in equally good performance with lower stiffness values. This might be due to perceptual interference, cognitive capabilities of processing multiple stimuli or through the occurrence of illusions. However, their device showed haptic applications are of value where stiffness discrimination is important.

### 4.4. Weight Exploration

The device from Mo et al. [25] described in Section 3.3 was also tested for rendering mass in a remote environment in combination with a touch HD. After a pre-training phase, the subjects needed to feel the mass of a remotely lifted box and compare it to other boxes with different weights and tell whether the first mass was lighter, heavier or the same as the second. The authors compared tests with only cutaneous feedback from the fingertip device to tests in which kinaesthetic feedback was also provided by the touch HD. Cutaneous feedback was slightly superior to kinaesthetic feedback in displaying weak stimuli and kinaesthetic feedback was slightly superior to cutaneous feedback in displaying strong stimuli. In conclusion, the fingertip device showed the capability of displaying mass information in a remote environment.

### 4.5. Shape Exploration 

Martinez et al. [28] designed a vibrotactile glove to identify virtual 3D geometric shapes. The glove contained 12 ERM actuators that were driven upon contact with the virtual object contours through pulse and pulse overdrive to reduce the latency. When a collision was detected, a vibration pulse, proportional to the speed and the angle of impact, was generated to simulate the surface impact. Sixteen participants were asked to identify virtual 3D shapes as fast and accurately as possible with the whole hand after they explored real paper models with the same dimensions. The number of successful identifications was far above the chance level. The results for identifying the cone shape showed a higher success rate, probably due to its morphologically different shape compared to other shapes. The fact that there was no force feedback made the task more complex and required additional concentration during exploration of the shapes. An average identification rate of 65% validated the device’s use in identifying virtual shapes. Longer training could improve the success rates.

### 4.6. Orientation Exploration

Hinchet et al. [30] developed a glove for precise manipulation of objects in VR that integrated both kinaesthetic and cutaneous feedback to the index finger and thumb, called DextrES. An electrostatic brake created force feedback up to 20 N of force at 1500 V due to the frictional forces that occurred between steel strips after an electrostatic force was generated between the strips. At the beginning of a virtual grasp, piezo actuators provided vibrotactile feedback to the fingertips to indicate the start of a touch event. Participants had to perform four different grasps, a lateral, a parallel, a power, and a pinch grasp in a VE, as accurately as possible with the different feedback modes: kinaesthetic force feedback, vibrotactile or combined. The results showed no difference in completion times of the different grasps for the different modes, but the combination of both feedback modes improved the precision of the grasps, especially for lateral, parallel and pincher grasps. A combination of both the piezo actuators and the electrostatic break seemed to improve the immersion and precision the most, with kinaesthetic only and then tactile only rated below this. 

Lee et al. [29] designed a cutaneous HD for the fingertip in combination with a novel finger-tracking module that can display 3-DoF contact forces to the fingertip. The device consists of a plate with three springs on the fingertip. Micro-DC motors generate the tensile forces on the fingertip through rigidly connected wires which are controlled by proportional integral control. During 16 virtual manipulation tasks, 10 participants had to insert a breakable peg into a horizontally placed hole. The participants were allowed to practice with the device before completing the tasks. The results indicated that cutaneous feedback provided faster task completion times, and thus better performance, than without. 

The fingertip device from Mo et al. [25], as discussed in both Section 4.3 and Section 4.4, was also tested during a pick and place task. The participants had to pick up a virtual cube with a certain orientation that could be “crushed” when too high forces were applied during the grasp. The results showed that the cutaneous feedback improved the performance of the task, especially when two fingers received the tactile force feedback.

## 5. Discussion

### 5.1. General

This overview shows that virtual exploration of OP has been tested in VR with a variety of wearable haptic technologies. Recognizing textures or materials of objects in VR is mostly related to the tactile perception of the human skin; the devices from Keef et al., 2020 [22] and Yem et al., 2017 [10], concerning texture exploration, both focus on tactile perception. They use combinations of electrotactile and vibrational feedback to simulate surface hardness and roughness. Surfaces with roughness at the macro- or micro-scale are perceived by multiple types of mechanoreceptors in the human skin. This requires an HD to address both the high frequency receptors for micro roughness and the low frequency receptors and pressure and shear receptors for larger structures. The combination of electrotactile and vibrational feedback seems sufficient for simulation of micro roughness, but, for surfaces with larger features, actual indentation of the fingertip could be necessary. Gabardi et al., 2016 [31] constructed a wearable fingertip device for texture exploration that can give three DOF cutaneous force feedback by means of a kinematic configuration connected to a smooth plate actuated by two servo actuators for the shear forces, and a voice coil actuator to actuate the contact force and high frequency vibrations in combination with a tele-operated kinaesthetic probe. In their study, the participants had to explore the edges and textures of a surface with kinaesthetic feedback by means of the probe and/or cutaneous feedback through the tactile device. The tactile sensory information proved to be crucial for completing the perception task compared to kinaesthetic feedback only, underlining the importance of cutaneous feedback in texture exploration. Thermal quality exploration of objects in VR has been tested through the combination of ERM actuators with semi-conducting refrigerators. Both hot and cold temperatures are distinguished by participants, but it seems difficult to convey very detailed thermal information, partly due to heat-management issues. Interestingly, training beforehand with the system does not affect participant performance [23], suggesting that the thermal quality is a characteristic that feels more natural compared to, for example, texture through mechanical feedback, where training does play a role in performance. 

The exploration of object compliance has attracted much interest in the field of haptic virtual object interfaces [13,32]. This is possibly due to the importance of stiffness in determining the applied forces needed to successfully grasp objects and indicates the importance of object compliance exploration in VR. We reviewed several HDs which address either tactile or kinaesthetic perception to convey stiffness information [24,25,26,27]. The virtual stiffness is generally expressed through a force or vibration intensity proportional to the forces that naturally occur from interacting with a spring. Tendon-based kinaesthetic mechanisms seem promising due to their light and flexible nature, making the mechanism compact and easily adjustable to different hand sizes. However, as with all force-feedback devices, the reaction forces that occur due to the feedback must be counteracted at other locations, mostly the wrist, interfering with the perception of the object interaction. The device described by Lee et al., 2019 [29] can only provide three different stiffness modes because of a selective locking mechanism, whereas the electrostatic braking mechanism of Hinchet et al., 2018 [30] provides more continuous modes of feedback by adjusting the braking forces. They also found that kinaesthetic feedback scored better in terms of realism compared to vibrotactile feedback. Nevertheless, kinaesthetic mechanisms require more power to counteract the forces from the fingers in comparison to the use of vibrational actuators for tactile stimulation only, making kinaesthetic devices less suitable for potential wireless application. The device described by Maereg et al. (2017) [27], for example, is capable of conveying stiffness information by using only vibrational actuators in a wireless device, which increases the wearability, range of motion and ease of use of such haptic systems. In general, for both tactile and kinesthetic perception, higher virtual stiffnesses seem easier to distinguish and compare to real world objects. Moreover, it was shown that accurate concurrent visual feedback plays an important role in the implementation of haptic feedback for stiffness exploration [26], meaning that, in order to convey proper haptic information, the corresponding visual feedback must match the haptic feedback. 

Weight is a very complex object characteristic to simulate. Gravity acts upon objects, creating moments with respect to the user’s body. This entails mostly kinaesthetic perception, but also some tactile aspects due to friction and fingertip pressures. Therefore, the combination of both tactile and kinaesthetic feedback is important in mimicking the weight of an object in VR. The discussed tactile feedback device designed by Mo et al. (2019) [25] shows that tactile feedback adds to the perception of weight compared to kinaesthetic feedback only and that it is possible to convey virtual weight information through haptics [33]. The handheld HD by Choi et al. (2017) [16] can convey weight information in a virtual environment; however, highly wearable tactile feedback in combination with kinaesthetic feedback for weight exploration in VR has, as far as we know, not been tested and remains an untouched field.

The HD of Martinez et al. (2016) [28] that was user-tested for virtual shape exploration uses ERM actuators that provide vibrational feedback proportional to the speed and angle of impact of the object. The results showed that it is possible to convey shape information though vibrations; however, participant training is necessary to improve performance as the lack of kinaesthetic feedback requires more concentration during the exploration task. Therefore, it would be interesting to research the possibilities of shape exploration with added kinaesthetic feedback further in a more wearable setup. 

In general, vibrational actuators show high flexibility in simulating different types of object characteristics and are a very wearable, low-cost and low-energy consuming solution for haptic feedback. Vibrational actuators are now available in many sizes, shapes, and types and, with the right control schemes, they provide low latency feedback. However, vibrations are usually perceived as less realistic compared to force feedback devices, because vibrations are not regularly encountered by users when manipulating objects in real life, except for, e.g., using an electric toothbrush, or holding a steering wheel during driving. Fingertip devices that provide force feedback simulate a more realistic sense of pressure on the fingertips, but they are still relatively bulky and can restrict the range of motion of the fingers [25], [29]. Compared to this, tendon-driven mechanisms are a more compact and lightweight solution to simulate kinaesthetic force feedback [24,26]. 

A combination of both tactile and kinaesthetic feedback seems most realistic for haptic object interaction in general. In particular, the versatility of vibrational actuators in combination with tendon-driven mechanisms shows potential across multiple OP exploration tasks. However, interference of multiple different stimuli and perceptual masking can reduce the effectiveness of the feedback and the performance of users [27]. Some studies have been performed without visual feedback to fully focus on the perception of the haptic feedback. However, when visual feedback was used, it was shown that providing fitting visual feedback had a great influence on the perception and performance of the object interaction [24]. This suggests that, to create realistic VR experiences, all the feedback modalities must be synchronized with minimal latency between the different modalities. The lack or implementation of other feedback modalities, such as visual and auditory feedback, could have an impact on usability [4].

Another factor that influences the outcomes of user tests with HDs could be learning effects that may emerge after multiple attempts at an interaction [31]. Most studies try to eliminate this by familiarization and training with the device. However, there are differences in how, and to what extent, the participants were trained before the experiments, and there is still a chance that learning throughout the experiments affected the data. Control tests were not performed to determine the optimal training period, making it hard to verify whether a participant can be marked as ‘trained’ and to eliminate learning biases from the results.

Additionally, the different studies showed large variability in the testing methods used. Both the experimental setups and the questionnaires differed across studies, making it hard to compare outcomes. Together with the small subject pools and differences in age of subjects in different studies, caution is warranted when drawing conclusions from the results of VR experiments. It is possible that younger participants have been more exposed to games and VR in general as opposed to older participants, resulting in a possible bias in studies. Moreover, some studies did not include details of the gender, age, and hand-preference of participants [22].

### 5.2. Future Recommendations

The combination of both tactile and kinaesthetic feedback has been shown to be most realistic and new technologies should focus on combining these into highly wearable devices that can address all OP exploration modalities. Active and soft materials, such as polymer-based materials, have the potential to allow for a more realistic and broader spectrum of tactile sensations due to their light and flexible properties [19,34]. Implementing or combining sensors into actuators could also increase wearability and allow for more real-time feedback [35]. This synchronization of the visual and haptic feedback during the VR tasks is important for realism and immersion [24,26,27]. Combined smart-sensing and actuating mechanisms, together with improved VR rendering algorithms, could, therefore, enhance haptic perceptions and interaction during virtual object exploration [36]. Kinaesthetic solutions will provide a bigger challenge due to the larger forces needed. Many force-feedback devices are, therefore, exoskeletons [37,38,39], and further optimization and minimization of TSAs is necessary [40]. The step from new emerging haptic technologies to actual VR implementation plays a key role in their performance. Due to the large variability in testing methods, there is a need for a validated evaluation method to properly evaluate and compare VR HDs when tested in a VE. This method should include several aspects of the overall haptic experience with the HD, such as wearability, ease of use and realism of haptic feedback. As far as we know, no model has yet been validated for this purpose, but models such as the HX model seem to represent promising solutions to evaluate when validated for use [41]. Moreover, the method should have a generalized comparable outcome and should be applicable to physically testing a broad variety of HDs. Here, VR evaluation based on performance in all the different OP exploration modalities could be of great use. This classification provides a clear structure for object interaction, which represents an important means of interacting with our environment. This evaluation could be in the shape of a standardized test with virtual objects for which the properties can be easily changed. For OP compliance, texture, weight and thermal quality, the test can be based on grading the different objects, for example, from, very stiff to not stiff, very rough texture to very fine texture, very cold to very warm, etc. Tracking the time it takes to complete the task and the accuracy of the users can then provide highly quantifiable results. For the OP shape and orientation, the test can be based on the game of putting a specific shape in a corresponding hole. Here, completion time and a measure of motion smoothness, such as the minimum jerk model or spectral arc length [42], can provide quantifiable results. The VR test should be fully virtual, leaving no room for differences when comparing to physical objects. Moreover, the test can be performed both with visual feedback and without to identify the feedback inconsistencies in the system [41].

## 6. Conclusions

This work provides an overview of recently developed wearable HDs that are categorized according to their OP exploration modalities (i.e., texture, thermal quality, compliance, weight, size/shape, and orientation) and that have been verified in user experiments. Different technological solutions and combinations were found for the different exploration tasks. 

The lack of general testing methods for object exploration makes performance comparison of the different technologies complex and stresses the need for a generalized testing model. This review shows that a combination of tactile and kinaesthetic feedback, specifically vibrotactile and tendon-driven mechanisms, has the potential for more extensive and detailed OP exploration. VR will especially benefit when new materials combining sensing and actuation are implemented and the devices are minimized and optimized in terms of feedback complexity.

## Figures and Tables

**Figure 1 sensors-23-01563-f001:**
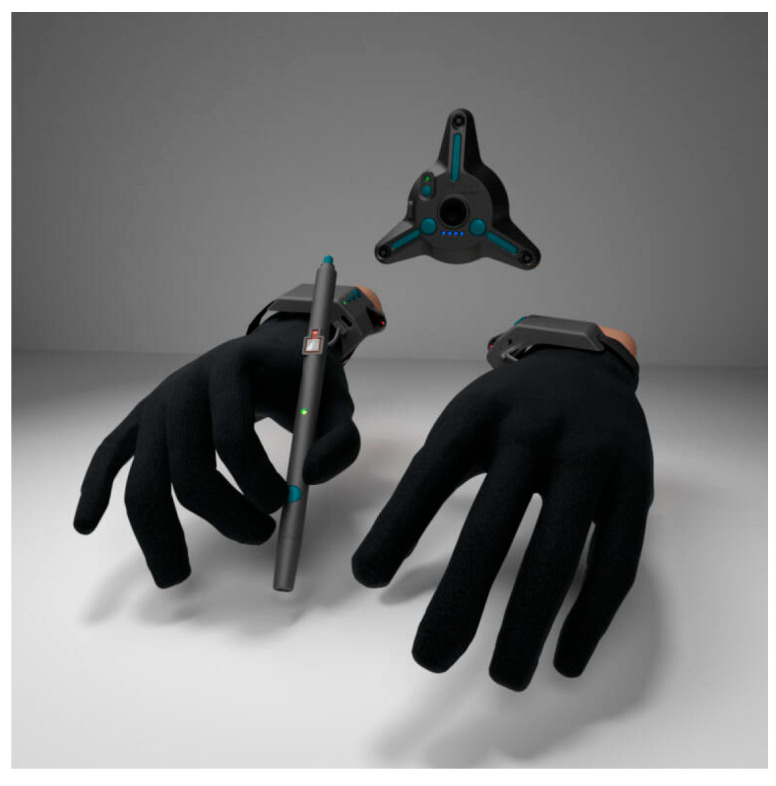
VRfree^®^ system (Glove, Stylus and Headmodule) by Sensoryx AG.

**Figure 2 sensors-23-01563-f002:**
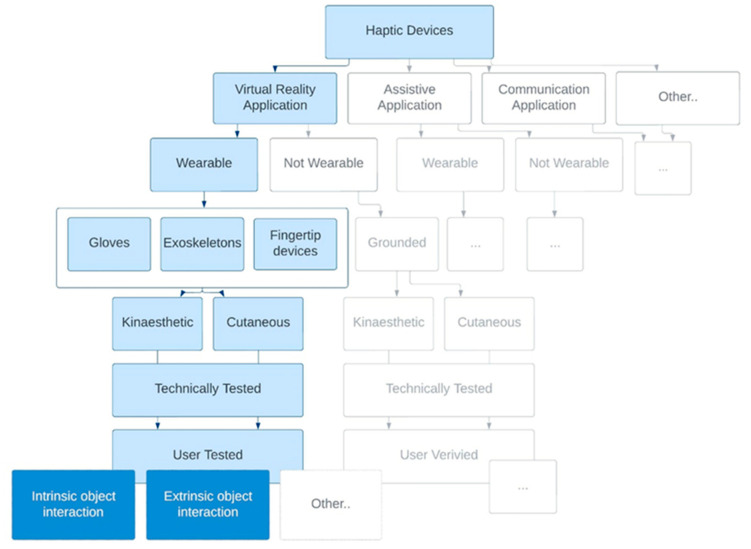
Overview of scope of HDs included in review.

**Figure 3 sensors-23-01563-f003:**
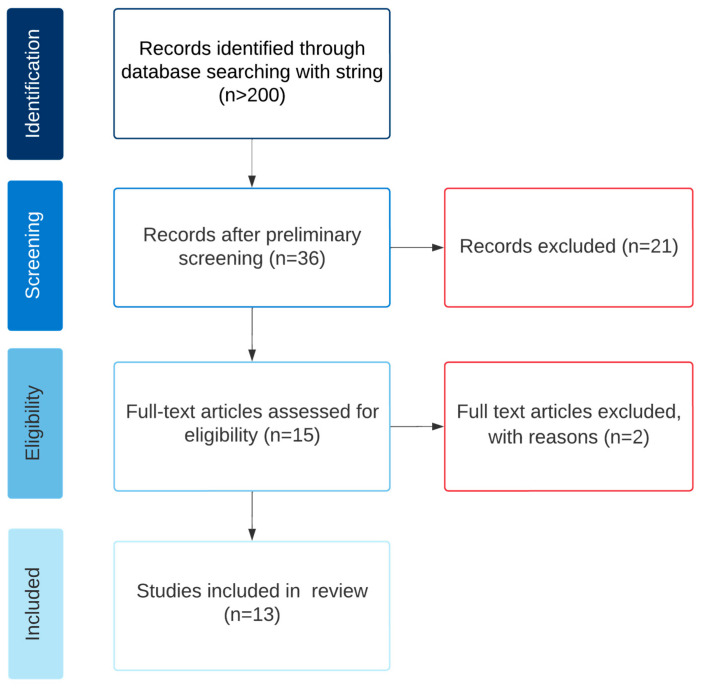
Flowchart of article selection.

**Table 1 sensors-23-01563-t001:** Overview of haptic wearable devices tested on virtual object property exploration.

Reference	Function	Type of Feedback	Haptic Technology	Tracking Technology	Conclusion
Keef et al., 2020 [22]	Texture	Cutaneous	Glove with electrotactile feedback through conductive polymer electrodes and vibrotactile feedback through ERM ^1^ actuators	Flex-sensors forfinger positionand HTC Vive headset with motion tracker for hand position	The device is capable of providing a sensation of roughness through electrotactile feedback
Yem et al., 2017 [10]	Texture	Cutaneous	DC ^2^ motor for mechanical (vibro)tactile stimulation in combination with an electrode film for electrical stimulation	Computer-mouse	The device is capable of providing four dimensions of tactile feedback in a virtual texture exploration task.
Li et al., 2020 [23]	Thermal Quality	Cutaneous	Miniature ERM ^1^ actuators and two semiconductor refrigerators	Keyboard	The device is capable of providing temperature feedback conforming to skin characteristics.
Keef et al., 2020 [22]	Thermal Quality	Cutaneous	Thermoelectric devices	Flex-sensors for finger positionand HTC Vive system with motion tracker for hand position	The device is capable of expressing thermal quality.
Lee et al., 2021 [24]	Compliance	Kinaesthetic	Tendon driven with a selective locking mechanism	Motion capture system with infra-red markers and cameras	The mechanism can render rigid and elastic/ deformable virtual objects.
Mo et al., 2019 [25]	Compliance	Cutaneous	Three small servos and an uncoupled fivebar and a slider-crank linkage combination to provide 3-DoF ^3^ force feedback	Leap motion	The devices proved to be capable of rendering variable stiffnesses and users not only use haptic cues but also visual cues in detecting spring stiffness differences.
Hosseini et al., 2018 [26]	Compliance	Kinaesthetic	TSA ^4^ through a DC ^2^ motor	HTC Vive system	TSA ^4^ is a compact, lightweight solution for haptic rendering of virtual stiffnesses.
Maereg et al., 2017 [27]	Compliance	Cutaneous	Vibrotactile feedback with ERM ^1^ actuators and haptic controllers	Oculus Rift and Leap Motion	The device is capable of rendering stiffness for virtual stiffness discrimination.
Mo et al., 2019 [25]	Weight	Cutaneous (/kinaesthetic)	Three small servo motors and an uncoupled fivebar and a slidercrank linkage combination to provide 3-DoF ^3^ force feedback and an external kinaesthetic device	Kinaesthetic device ”Touch Haptic”	The performance of the subjects with cutaneous feedback was close to that with kinesthetic feedback. The device was capable of displaying mass information during remote manipulation.
Martinez et al., 2016 [28]	Size/shape	Cutaneous	12 ERM ^1^ actuators, actuated with PWM ^5^ and pulse overdrive	Phase Space Impulse optical tracking system, which can track multiple LED markers at 480 Hz	The device achieves a good response rate in terms of shape identification, butalso has shortcomings due to the lack of kinaesthetic feedback.
Lee et al., 2019 [29]	Orientation	Cutaneous	Fingertip device with three DC ^2^ motors	A finger-tracking module with IMUs ^6^ and soft sensors	The importance of haptic feedback and full motion tracking is verified together with the performance of the device to enhance task completion time
Mo et al., 2019 [25]	Orientation	Cutaneous	Three small servo motors and an uncoupled fivebar and a slidercrank linkage combination to provide 3-DoF ^3^ force feedback	Leap Motion	The device with cutaneous feedback improved task performance
Hinchet et al., 2018 [30]	Orientation	Cutaneous/ kinaesthetic	Thin electrostatic brake for force feedback and piezo actuators for vibrotactile feedback	Optical tracking system	The device is capable of improving grasping precision through force and vibrotactile feedback.

^1^ Eccentric Rotating Mass (ERM), ^2^ Direct Current (DC), ^3^ Degrees of Freedom (DoF), ^4^ Twisted String Actuation (TSA), ^5^ Pulse Width Modulation (PWM), ^6^ Inertial Measurement Unit (IMU).

## Data Availability

Not applicable.

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
