# Peer review of "An Overview of Wearable Haptic Technologies and Their Performance in Virtual Object Exploration"

_sensors, 2023, doi:10.3390/s23031563_

Round 1
Reviewer 1 Report
The research problem is good but the way of presenting the same would not reflect the quality of the work as same type of works are already present.
As it is very general review type work so some bibliometric study needs to be incorporated and different types of tabular representations and graphical representations should be present in your work which is missing.
Flowchart of article selection is good , but if it is automated then only some quality reflection should be there.
Technological algorithmic study must be there which is also needed to be incorporated.
Basic objective of your work must be revised and how ARVR is changing the world in the form of view it should be clearly explained.
Author may check this work and its reff : https://onlinelibrary.wiley.com/doi/10.1002/adfm.202008831
Author Response
Reviewer 1:
Comment 1: The research problem is good but the way of presenting the same would not reflect the quality of the work as same type of works are already present.
Response 1: Many thanks for your input. We now specifically write in the introduction section what is new about the review. Also following Reviewer 3, we now included a short State of the Art chapter.
Comment 2: As it is very general review type work so some bibliometric study needs to be incorporated and different types of tabular representations and graphical representations should be present in your work which is missing.
Response 2: We further included some References (New References 34 to 41) and also followed the Reviewer’s suggestion and added, https://onlinelibrary.wiley.com/doi/10.1002/adfm.202008831 (Reference 34). We did not add additional tabular and graphical representations since the focus of this review was to discuss the different technological solutions for different object properties exploration, useful for the design of future haptic object interactions in virtual/augmented reality.
Comment 3: Flowchart of article selection is good, but if it is automated then only some quality reflection should be there.
Response 3: We revised the Flowchart and did a quality check.
Comment 4: Technological algorithmic study must be there which is also needed to be incorporated.
Response 4: We added more technical information in the discussion section (especially in the future recommendation section) related to technical aspects.
Comment 5: Basic objective of your work must be revised and how ARVR is changing the world in the form of view it should be clearly explained.
Response 5: We revised the aim of our work in the introduction section.
Reviewer 2 Report
Comments
1. The outcomes of the reviewing of wearable haptic devices categorized by the OP exploration must be briefly included in the abstract. In addition to this, the recommendations for the future scope must be included.
2. The authors must focus on sentence formation from the perspective of research writing. In the introduction section, the authors started a sentence with “Where computer-generated simulations of three-dimensional environments sounded far away some years back, VR has recently emerged in a wide variety of applications”. It is not acceptable to start a sentence this like in the research article.
3. The contribution and organization of the study must be included at the end of the introduction section.
4. The explanation of table 1 must be discussed in detail, as it contains various aspects that must also be addressed in paragraph form.
5. A comparative table must be included in the study to justify the novelty of this review when compared with the previous studies.
Author Response
Reviewer 2:
Comment 1: The outcomes of the reviewing of wearable haptic devices categorized by the OP exploration must be briefly included in the abstract. In addition to this, the recommendations for the future scope must be included.
Response 1: Many thanks for this suggestion. We now included the outcomes and also recommendations in the abstract.
Comment 2: The authors must focus on sentence formation from the perspective of research writing. In the introduction section, the authors started a sentence with “Where computer-generated simulations of three-dimensional environments sounded far away some years back, VR has recently emerged in a wide variety of applications”. It is not acceptable to start a sentence this like in the research article.
Response 2: We agree and revised this sentence.
Comment 3: The contribution and organization of the study must be included at the end of the introduction section.
Response 3: We agree and now added more info (in the introduction and added a new section called state-of-the art/aim) what is exactly the contribution of this review.
Comment 4: The explanation of table 1 must be discussed in detail, as it contains various aspects that must also be addressed in paragraph form.
Response 4: We further explained the table in the beginning of the results section. The contents of the table are then discussed paragraph.
Comment 5: A comparative table must be included in the study to justify the novelty of this review when compared with the previous studies.
Response 5: Many thanks for your suggestion. We now specifically write at the end of the introduction, and also in a new state-of-the-art section, what is the novelty of this review.
Reviewer 3 Report
The paper is written clearly, and the topic is very interesting and up-to-date. The abstract is concise and clear. The problem description and research goals are sound and clear. The discussion presents the differences between the analyzed technologies as well as their pros and cons. The conclusion points towards possible research gaps in this field.
However, the paper cannot be accepted in its present form. In order to improve its quality, the following remarques have to be resolved:
1. keywords:
The word INTERACTION is too general, a more precise term (OBJECT INTERACTION) would be better
The word SYSTEM is also too general, HAPTIC SYSTEM or VR SYSTEM would be better
2. Separate STATE OF THE ART from INTRODUCTION. State of the art is a key chapter in the review article and it is missing here. The structure of the paper is not given.
3. There are not enough references for a review paper!
Considering constant development and improvements in VR technologies, a change of search procedures and keywords is needed. Improper searching has led to a small number of analyzed references. Why was WEB OF SCIENCE not included in the search as it has a considerable pool of articles?
The main keywords we used for the search strategy were: Haptic, cutaneous, tactile, kinaesthetic, force feedback AND object, interaction, shape, stiffness, texture, weight, thermal, orientation AND virtual, augmented reality AND wearable, glove, exoskeleton, fingertip device, AND subjects, volunteers, participants, users.
A search with this many keywords has been misleading. Using less keywords, statements or combinations of the keywords would result in more relevant results.
For instance, using “haptic devices for vr” gave the following reference (review article):
Haptic display for virtual reality: progress and challenges, Virtual Reality & Intelligent Hardware,Volume 1, Issue 2, April 2019, Pages 136-162, DangxiaoWANG, YuanGUO, ShiyiLIU, YuruZHANG, WeiliangXU, JingXIAO
4. The font in Figs. 2 and 3 is not clearly visible. Use a larger font or use higher resolution for Figs. (if that is the problem)
5. Reference 5 is incomplete: date, journal/conference name, doi missing
Reference 8 is incomplete: date, doi missing
Reference 12 is incomplete: date, journal/conference name, doi missing
Author Response
Reviewer 3:
The paper is written clearly, and the topic is very interesting and up-to-date. The abstract is concise and clear. The problem description and research goals are sound and clear. The discussion presents the differences between the analyzed technologies as well as their pros and cons. The conclusion points towards possible research gaps in this field
Response: Many thanks for these kind words
However, the paper cannot be accepted in its present form. In order to improve its quality, the following remarques have to be resolved:
Comment 1: keywords: The word INTERACTION is too general, a more precise term (OBJECT INTERACTION) would be better
The word SYSTEM is also too general, HAPTIC SYSTEM or VR SYSTEM would be better
Response 1: Many thanks for your good suggestions. We revised the key words and followed your inputs.
Comment 2: Separate STATE OF THE ART from INTRODUCTION. State of the art is a key chapter in the review article and it is missing here. The structure of the paper is not given.
Response 2: We have reviewed Sensors authoring guidelines and according to these Sensors do not require a STATE OF THE ART chapter. However, after contacting the managing editor we now include a short chapter called «State of the art/aim (new chapter 2)
Comment 3: There are not enough references for a review paper! Considering constant development and improvements in VR technologies, a change of search procedures and keywords is needed. Improper searching has led to a small number of analyzed references. Why was WEB OF SCIENCE not included in the search as it has a considerable pool of articles?
The main keywords we used for the search strategy were: Haptic, cutaneous, tactile, kinaesthetic, force feedback AND object, interaction, shape, stiffness, texture, weight, thermal, orientation AND virtual, augmented reality AND wearable, glove, exoskeleton, fingertip device, AND subjects, volunteers, participants, users.
A search with this many keywords has been misleading. Using less keywords, statements or combinations of the keywords would result in more relevant results.
For instance, using “haptic devices for vr” gave the following reference (review article):
Haptic display for virtual reality: progress and challenges, Virtual Reality & Intelligent Hardware,Volume 1, Issue 2, April 2019, Pages 136-162, DangxiaoWANG, YuanGUO, ShiyiLIU, YuruZHANG, WeiliangXU, JingXIAO
Response 3: Many thanks for your suggestion. We now added more references and also added the suggestion of the reviewer. We used Pubmed and IEEE Explorer which are the biggest databases. The aim of the present review was outlined in detail in the introduction, and following this aim we specified our key word search. We agree that this might be a predefined selection, however on the other hand we avoid another general review about VR/AR. These reviews already exist. Finally, we believe that the quality of a review is not related to the quantity of articles, but to the inclusion of the articles that address the review's specific objective.
Comment 4. The font in Figs. 2 and 3 is not clearly visible. Use a larger font or use higher resolution for Figs. (if that is the problem)
Response 4: Many thanks for this good suggestion. We revised the font in Figs 2 and 3.
Comment 5: Reference 5 is incomplete: date, journal/conference name, doi missing. Reference 8 is incomplete: date, doi missing. Reference 12 is incomplete: date, journal/conference name, doi missing
Response 5: We have now revised all references and these should now be complete.
Round 2
Reviewer 3 Report
Paper is significantly improved.
Authors have noticed large differences in testing methods which make evaluation and comparison of the test results very difficult. Are lines 450-456 in Future Recommendations related to that problem? It yes then please clarify it, if not then it would be beneficial to suggest ideas for future work.
Author Response
Comment 1:
Paper is significantly improved.
Authors have noticed large differences in testing methods which make evaluation and comparison of the test results very difficult. Are lines 450-456 in Future Recommendations related to that problem? It yes then please clarify it, if not then it would be beneficial to suggest ideas for future work.
Response 1: Many thanks for this fine input. We further revised these lines in the Future recommendations and now write the following: «Due to the large variability in testing methods, there is need for a validated evaluation method to properly evaluate and compare the VR haptic devices when tested in a VE. This method should include several aspects of the overall haptic experience with the HD, such as wearability, ease of use and realism of haptic feedback. As far as we know, no model has yet been validated for this purpose, but a model, such as the HX model, seems a promising solution to evaluate when validated for use [41]. Next to that, the method should have a generalized comparable outcome and it should be applicable to physically test a broad variety of HDs. Here, VR evaluation based on performance in all different OP exploration modalities could be of great use. This classification provides a clear structure to object interaction, which is an important means of interacting with our environment. This evaluation could be in the shape of a standardized test with virtual objects of which the properties can be easily changed. For OP compliance, texture, weight and thermal quality, the test can be based on grading the different objects, for example, from, very stiff to not stiff, very rough texture to very fine texture, very cold to very warm, etc. Tracking the time it takes to complete the task and the accuracy of the users can then provide very quantifiable results. For the OP shape and orientation, the test can be based on the game of putting a specific shape in a corresponding hole. Here, completion time and a measure of motion smoothness, such as the minimum jerk model or spectral arc length [42], can provide quantifiable results. The VR test should be fully virtual, leaving no room for differences when comparing to physical objects. Also, the test can be performed both with visual feedback or without, to identify the feedback inconsistencies in the system [41].
We added also a new reference 42: S. Balasubramanian, A. Melendez-Calderon, A. Roby-Brami, and E. Burdet, “On the analysis of movement smoothness,” Journal of Neuro Engineering and Rehabilitation, vol. 12, p. 112, Dec. 2015, doi: 10.1186/10.1186/s12984-015-0090-9.